# Effects of Osiris9a on Silk Properties in *Bombyx mori* Determined by Transgenic Overexpression

**DOI:** 10.3390/ijms21051888

**Published:** 2020-03-10

**Authors:** Tingcai Cheng, Xia Zhang, Zhangchuan Peng, Yinfeng Fan, Lin Zhang, Chun Liu

**Affiliations:** State Key Laboratory of Silkworm Genome Biology, Southwest University, Chongqing 400716, China; chengtc@swu.edu.cn (T.C.); zxzxzx919191@163.com (X.Z.); sqdpzc@163.com (Z.P.); Fan_Yinfeng@163.com (Y.F.); zhanglincqbb@163.com (L.Z.)

**Keywords:** Osiris, transgenic overexpression, sericin, mechanical properties, silk, silkworm

## Abstract

Osiris is an insect-specific gene family with multiple biological roles in development, phenotypic polymorphism, and protection. In the silkworm, we have previously identified twenty-five Osiris genes with high evolutionary conservation and remarkable synteny among several insects. *Bombxy mori*
*Osiris9a* (*BmOsi9a*) is expressed only in the silk gland, particularly in the middle silk gland (MSG). However, the biological function of BmOsi9a is still unknown. In this study, we overexpressed *BmOsi9a* in the silk gland by germline transgene expression. *BmOsi9a* was overexpressed not only in the MSG but also in the posterior silk gland (PSG). Interestingly, *BmOsi9a* could be secreted into the lumen in the MSG but not in the PSG. In the silk fiber, overexpressed *BmOsi9a* interacted with Sericin1 in the MSG, as confirmed by a co-immunoprecipitation assay. The overexpression of *BmOsi9a* altered the secondary structure and crystallinity of the silk fiber, thereby changing the mechanical properties. These results provide insight into the mechanisms underlying silk proteins secretion and silk fiber formation.

## 1. Introduction

The Osiris gene family, first described in *Drosophila melanogaster*, is a large conserved family in insects [1,2]. Twenty-four Osiris genes have been identified in the *D. melanogaster* genome, including 20 that are clustered on chromosome 3R within a 168-kb region, which is both triplo-lethal and haplo-lethal [1,3]. The knockdown of either *Osiris6 (Osi6)* or *Osi7* during embryonic development results in nearly complete lethality [4]. *Osi6*, *Osi7*, and *Osi8* can alter octanoic acid resistance in adult *D. melanogaster* and contribute to octanoic acid resistance in *D. sechellia* [4,5,6]. *Osi21,* also known as *diehard4,* inhibits the recycling of rhodopsin in the retina as part of the endocytic pathway [7]. Seven Osiris proteins are highly expressed in vesicles at and near the apical membrane in the trachea, suggesting that they are likely involved in tube maturation via vesicular trafficking or interactions with other apical membrane proteins [8]. *Osiris23,* also known as *gore-tex,* encodes an endosomal protein that is essential for envelope curvature, nanopore formation, and odor receptivity and is specifically expressed in developing olfactory trichogen cells [9]. Osiris genes are involved in the immune response in the honey bee [10] and in wing development in *Bombxy mori* [11,12]. The gene family has conserved biological roles in insect development, phenotypic polymorphism, and protection [13].

In the silkworm, twenty-five Osiris genes have been identified, and 20 are located on chromosome 26 in a cluster within a 150-kb region. Most of these genes showed wing-specific or epidermis-specific expression [11]. A serial analysis of gene expression has shown that one member of the osiris9 subfamily is exclusively expressed in the middle silk gland (MSG) but not in the posterior silk gland (PSG) [14]. Our previous study also confirmed that *Osiris9* (*BmOsi9a*) is highly expressed only in the MSG and BmOsi9a is secreted into the sericin layer [15]. Moreover, Osi9a is evolutionarily conserved and is a common component of silk fibers in silk-producing Bombycidae and Saturniidae insects [15]. However, the biological function of Osi9a in silk fibers is still unclear.

*B. mori* cocoon silk is mainly composed of fibroin and sericin proteins [16]. The mechanical properties of silk at different developmental stages differ significantly and are related to the silk components [17]. BmOsi9a was detected in scaffold silk and cocoon silk [18]. The BmOsi9a content is approximately twice that of Sericin3 (Ser3) and half that of Sericin1 (Ser1) in cocoon silk [17], suggesting that BmOsi9a is a major silk component. However, it is not clear whether BmOsiris9a is related to the mechanical properties of silk.

To explore the biological function of *BmOsi9a* in the silk gland, in this study, we constructed a transgenic line with *BmOsi9a* overexpression in the silk gland for analyses of the mechanical properties and structures of silk. These results provide insight into the formation of silk fiber in the silkworm.

## 2. Results

### 2.1. Transgenic Overexpression of BmOsi9a in the Silk Gland

BmOsi9a is one of components of the cocoon silk, which is synthesized and secreted from the MSG [15]. To explore the biological function of *BmOsi9a* in the silk gland, a *piggyBac* transgenic vector was constructed, containing pBac[hSer1sp-BmOsi9a] and pBac[3xP3-EGFP] (Figure 1A). The Ser1 promoter is widely used to construct the transgenic system to highly express proteins in the MSG [19,20]. The transgenic and helper plasmids were injected into 250 preblastoderm eggs; of these, 175 eggs hatched and developed to the adult stage. An EGFP-positive brood was obtained and used to establish the transgenic overexpression line (Figure 1B). Reverse PCR results showed that the insertion site was located in the intergenic region, 12739746–12739818 of scaffold 18 on chromosome 6.

The mRNA levels of exogenous *BmOsi9a*, *Myc*, and endogenous *BmOsi9a* in the MSG and PSG of day-3 fifth-instar larvae of the transgenic and wild-type lines were detected using qRT-PCR (Figure 1B). In the MSG, exogenous *Myc* levels were high in the transgenic line but not in the wild-type line (Figure 1B). The relative mRNA level of exogenous *BmOsi9a* was almost twice that of endogenous *BmOsi9a* in the wild-type line, indicating that *BmOsi9a* was overexpressed in the MSG. In the PSG, interestingly, exogenous *BmOsi9a* and *Myc* were also transcribed in the transgenic line, but endogenous *BmOsi9a* was not transcribed in the wild type (Figure 1B), consistent with our previous results [15].

To confirm whether exogenous BmOsi9a with a Myc-tag was synthesized in the transgenic line, proteins were extracted from the MSG and PSG of day-3 fifth-instar larvae (Figure 1C) for Western blotting. The expression of exogenous BmOsi9a was consistent with the results obtained at the mRNA level. In the wild-type line, BmOsi9a was detectable only in the MSG and not in the PSG (Figure 1C). Clear bands for Myc and BmOsi9a in the MSG of the transgenic line were found; their corresponding signal bands could be detected in the PSG of the transgenic line.

### 2.2. Location of BmOsi9a in the Silk Gland

According to our previous study, BmOsi9a is synthesized in the MSG, secreted into the lumen, and localized in the sericin layer [15]. To locate the overexpressed BmOsi9a, immunofluorescence was used to evaluate the silk glands of day-3 fifth-instar larvae. As determined by DAPI staining, nuclei were arranged in the cell layer of the MSG and close to the sericin layer. In the MSG of the wild-type line, only BmOsi9a could be detected in both the cell and the sericin layers, without green Myc fluorescence (Figure 2A). In the MSG of the transgenic line, the green Myc fluorescence signal could be detected in both the cell layer and the sericin layer; the red fluorescence signal of BmOsi9a was located in the same position and was brighter than that of the wild-type line, indicating that exogenous overexpressed BmOsi9a could be secreted into the lumen in the MSG (Figure 2A).

Exogenous BmOsi9a was detected at both transcriptional and translational levels in the PSG of the transgenic line, suggesting that *BmOsi9a* should be ectopically overexpressed (Figure 2B). BmOsi9a was not expressed in the PSG in the wild-type line. In the transgenic line, interestingly, both exogenous BmOsi9a and Myc could be observed in the cell layer but not in the lumen in the PSG (Figure 2B), indicating that exogenous BmOsi9a could not be secreted into the lumen in the PSG.

### 2.3. Interaction between BmOsi9a and Ser1

To detect whether exogenous BmOsi9a is secreted into cocoon silk, the proteins of cocoon silk were extracted for Western blotting. Fibrohexamerin/P25 fibroin was used as an internal reference protein. Bands corresponding to exogenous Myc were successfully detected in the cocoon silk of the transgenic line but not in the cocoon silk of the wild-type line (Figure 3A). Moreover, the BmOsi9a content was obviously higher in the transgenic overexpression cocoon silk than in the wild type cocoon silk. Interestingly, in addition to the signal band for BmOsi9a at 26 kDa, additional bands were observed at about 55 and 100 kDa, suggesting that BmOsi9a interacts with other proteins (Figure 3B). For identification of the interaction proteins, two bands were excised from SDS-PAGE gels for mass spectrometry analysis. Both results showed that the bands contained a component of Ser1, suggested that the proteins with low molecular weight might be Ser1 proteins translated by alternative splicing isoforms of *Ser1* gene. To verify the interaction between BmOsi9a and Ser1 by co-immunoprecipitation (Co-IP), proteins were extracted from the MSG, incubated with antibodies against BmOsi9a and BmSer1 bound to beads, and cross-linked. IgG was used as a negative control. Proteins were separated by SDS-PAGE and probed with antibodies. Among the BmOsi9a and BmSer1 Co-IP products, a product with a molecular mass of 26 kDa was detected by the BmOsi9a antibody (Figure 3C). In the MSG, BmOsi9a and BmSer1 were co-located in the cells and sericin layer (Figure 3D). These results suggest that BmOsi9a interacts with BmSer1 for secretion from cells to the lumen in the MSG.

### 2.4. Mechanical Properties of Silk

Cocoons of the transgenic and wild-type lines were harvested and single silk fibers were reeled from the cocoons. There were no obvious differences in the morphological properties of the cocoons and silk fibers between lines (Figure 4A). To determine the effect of BmOsi9a overexpression on the mechanical properties of silk fibers, mechanical tests were performed using single silk fibers, as described in our previous study [17,21]. The stress–strain curves for single silk fibers are shown in Figure 4B. By comparing the mean curves, we found that the silk fibers from the wild-type line were stronger than those from the transgenic line (Figure 4C). There were no significant differences in elongation and toughness between the wild-type and transgenic silks. The maximum strength of the wild-type silk was slightly greater than that of the silk of the transgenic line. The elastic modulus of the silk of the transgenic line was slightly higher than that of the wild-type silk. These results suggested that the overexpression of BmOsi9a changes the mechanical properties of the silk.

### 2.5. Secondary Structural and Crystal Morphological Characteristics of Silk Fibers

To explain the change in the mechanical properties of the silk fibers, the secondary structure of was detected by attenuated total reflectance Fourier-transform infrared spectroscopy (ATR-FTIR). The spectra of silk fibers were similar for the transgenic and wild-type lines with clear amide I, amide II, and amide III absorption bands in the wave number range of 1100–1800 cm^−1^ (Figure 5A). Secondary structural characteristics, including *β*-sheets and *α*-helix/random coils, are the commonly analyzed features of amide I and amide III bands [22,23]. The amide I band at 1580–1720 cm^−1^ was used for a peak deconvolution analysis (Figure 5A). According to previous reports, the deconvolution peak in the wave number range of 1615–1645 cm^−1^ and 1690–1700 cm^−1^ represent *β*-sheet structure, 1645–1665 cm^−1^ represents *α*-helix/random coils structure, 1680–1690 cm^−1^ represents *β*-turn structure [24,25,26,27,28,29]. The composition of secondary structures of the silk fibers differed between the transgenic and wild-type lines (Figure 5A). The peak (around 1610 cm^−1^) assigned to the *β*-sheet structure was slightly lower than that for the wild-type line. The *α*-helix/random coil content in the silk fiber was lower for the transgenic line (62.4 ± 5.4%) than for the wild-type line (56.4 ± 2.5%). The *β*-sheet content in the silk fiber was lower in the transgenic line (29.6 ± 1.6%) than in the wild-type line (37.1 ± 1.3%) (Figure 5A). We further evaluated whether the change in structural composition affected the crystallinity of silk fibers. By X-ray diffraction (XRD), the intensity of the crystalline peak and shape were very similar between the transgenic and wild-type lines (Figure 5B). The amorphous peak of the transgenic silk was slightly higher than that of the wild-type silk (Figure 5B). The crystallinity of the transgenic silk was approximately 49.38%, slightly lower than 54.66% for the wild-type silk (Figure 5B). These ATR-FTIR and XRD results indicated that the transgenic overexpression of BmOsi9a in the silk gland leads to changes in the secondary structure and thereby influences the mechanical properties of silk fibers.

## 3. Discussion

The Osiris gene family has multiple functions in insect biology. We previously found that *BmOsi9a* is the only gene in the family that is specifically expressed in the silk gland. BmOsi9a is synthesized and secreted into the lumen of the MSG and is localized in the sericin layer in the silk fiber [15]. In the present study, BmOsi9a was overexpressed in the MSG by germline transgene transmission using the *hr3*-linked promoter of *Sericin 1* (hSer1sp), which has been developed and optimized for the mass production of valuable recombinant proteins [19,20,30,31,32]. BmOsi9a was successfully overexpressed at the transcriptional and translational levels in the MSG of transgenic silkworms (Figure 1). The relative mRNA level of exogenous *BmOsi9a* was almost equal to the endogenous *BmOsi9a* level in the transgenic line. Additionally, *BmOsi9a* was ectopically overexpressed in the PSG in the transgenic line. We speculated there might be two reasons to explain the ectopic transgenic expression in a region-specific manner. One reason may be due to the position effect. Although the insertion position was located in the intergenic region, it also might cause a decrease in promoter specificity. Another explanation may be due to the *Ser1* promoter linked to the *hr3* enhancer, which changes a tissue-specific manner resulting in the ectopic expression of the exogenous gene in the PSG, as a previous report [30].

The silk gland of the silkworm is a specialized tissue; it includes the MSG for synthesizing and secreting sericin proteins and the PSG for synthesizing and secreting fibroin proteins [16]. Transgenic *BmOsi9a* could be translated and secreted into the lumen in the MSG (Figure 2A). Interestingly, although *BmOsi9a* could be ectopically overexpressed in the PSG, the protein could not be secreted into the lumen in the PSG (Figure 2B). This can be explained by the differences in the secretory mechanism between the MSG and the PSG. In the PSG, fibroin heavy chain, fibroin light chain, and P25 are assembled into a high-molecular-mass elementary unit for secretion into the lumen [33]. BmOsi9a was located in the sericin layer and interacted with Ser1, which is specifically expressed in the MSG (Figure 3). Thus, we speculated that BmOsi9a and Ser1 form a complex for secretion into the lumen in the MSG but that BmOsi9a could not be secreted without Ser1 in the PSG. However, further studies are needed to clarify the secretory mechanism of BmOsi9a.

The cocoon silk is composed of fibroin and sericin proteins, and the protein composition determines the structure and properties of silk [16]. The process of silk fiber formation is precise and complex. The molar ratio of three fibroin proteins (FibH, FibL, and P25) is approximately 6:6:1 in the cocoon silk [33]. Our previous work has indicated that the molar ratios and therefore the mechanical properties are significantly different among silks from different instars [17]. The content of BmOsi9a, a major silk component, was approximately twice that of Ser3 and half that of Ser1 [15,17]. The silk BmOsi9a content in the transgenic overexpression line was twice that in the wild-type line (Figure 1). The increase in the BmOsi9a content affected the mechanical properties and secondary structure of silk (Figure 4 and Figure 5). The overexpression of BmOsi9a decreased the crystallinity and *β*-sheet structures in the silk, which are determinants of the mechanical properties. These results suggest that BmOsi9a contributes to the formation of silk fibers with favorable mechanical performance. BmOsi9a as one of the components of cocoon silk that is mainly synthesized and secreted in the MSG, and can interact with Ser1 protein. Transgene overexpression of BmOsi9a in the PSG indicated that BmOsi9a was only synthesized and cannot be secreted into the lumen, suggesting that the secretory mechanism should be different between the MSG and PSG. It will provide a clue for the study of the synthesis and secretion mechanism of silk proteins in region-specific manner.

## 4. Materials and Methods

### 4.1. Insects

The nondiapausing silkworm strain *D9L* of *B. mori* was maintained at the State Key Laboratory of Silkworm Genome Biology, Chongqing, China, and was used for germline transformation. Eggs were maintained at 25 °C until hatching, and larvae were reared on fresh mulberry leaves.

### 4.2. Vector Construction

*BmOsi9a* was cloned from MSG cDNA using a forward primer with a *BamH*I restriction site and a reverse primer with a *Myc*-tag and *Not*I restriction site. The PCR product was digested with *BamH*I and *Not*I and inserted into the pSL1180 [hSer1sp-DsRed-Sv40] vector to replace the *DsRed* gene. The hSer1sp constructed by a previous study contains a *hr3* enhancer and the *Ser1* promoter with about 670 bp from the −518 to +141 relative to the transcription initiation site [20]. Finally, the hSer1sp-BmOsi9a-Sv40 cassette was digested with *Asc*I and inserted into the pBac [3xp3-EGFPaf] basic transgenic vector [34] to generate the transgenic vector pBac [hSer1sp-BmOsi9a-Sv40, 3xp3-EGFPaf].

### 4.3. Germline Transformation of the Silkworm

The purified DNA of the transgenic plasmid and a *piggyBac* transposase-expressing plasmid (pHA3PIG helper) [35], both at 500 µg/µL, were mixed and injected into the preblastoderm embryos 1–3 h after oviposition. The hatched G0 individuals were reared until the moth stage and mated with wild-type (WT) moths to generate G1 progeny. The G1 moths were screened for EGFP expression using a fluorescence stereomicroscope (Olympus, Tokyo, Japan). EGFP-positive G1 moths were backcrossed to generate G2 offspring for further analysis.

### 4.4. Quantitative Real-Time PCR

Total RNAs of the MSG and the PSG of WT and transgenic lines were extracted using TRIzol reagent (Invitrogen, Carlsbad, CA, USA) according to the protocol provided by the manufacturer. For reverse transcription, the PrimeScript RT Reagent Kit (Takara, Kusatsu, Japan) was used according to the manufacturer’s instructions. Quantitative RT-PCR (qRT-PCR) was conducted using the ABI7500 Real-time PCR Machine (Applied Biosystems, Foster City, CA, USA) and FastStar Universal SYBR Green Master Mix (Roche, Basel, Switzerland). Each qRT-PCR was performed under the following conditions: denaturation at 95 °C for 10 min, followed by 40 cycles at 95 °C for 10 s, 60 °C for 30 s, and 72 °C for 35 s. The expression of the target gene was calculated by Ct values. The mRNA levels were analyzed by qRT-PCR using specific primers for *BmOsi9a* (F: 5′-GCTGAAAACTTGCTCCGATGA-3′ and R: 5′-TTGATTCTCCCTGGCTCTGG-3′) and *Myc* (F: 5′-ATACTCCTCAAGAAACTGCTTTCC-3′ and R: 5′-TAGAGTCGCGGCCGCTTACAGATC-3′). *B. mori sw22934* was used as an internal control. All experiments were performed in triplicate and repeated three independently.

### 4.5. Western Blotting

The MSG and PSG were collected from Day-3 fifth instar silkworm larvae of the transgenic and wild-type lines. The silk proteins from cocoons were extracted as described in a previous study [18]. The tissues were homogenized in RIPA Lysis Buffer (Beyotime, Shanghai, China), which included 50 mM Tris (pH 7.4), 150 mM NaCl, 1% Triton X-100, 1% sodium deoxycholate, 0.1% SDS, and protease inhibitors. The supernatant from the homogenates was collected by centrifugation (10,000× *g* at 4 °C for 10 min). The protein concentrations were measured using a BCA Protein Assay Kit (Beyotime) using bovine serum albumin as a standard. The same amount of proteins (40 μg per well) was resolved by 12% SDS-PAGE and transferred onto polyvinylidene difluoride (PVDF) membranes (Roche). The membranes were blocked with 5% skim milk overnight at 4 °C and incubated with primary antibodies against BmOsi9a or Myc-tag (Thermo Fisher Scientific, Waltham, MA, USA) for 2 h at 37 °C. The antibody of BmOsi9a was obtained from our previous study [15]. After washing, the membranes were incubated with goat anti-rabbit IgG labeled with horseradish peroxidase (HRP) (Sigma, St. Louis, MO, USA) secondary antibodies. The protein bands were visualized with SuperSignal West Femto Maximum Sensitivity Substrate (Thermo Scientific, Waltham, MA, USA) using the automatic exposing pattern of Genome XRQ (Gene Company, Hong Kong).

The 55-kDa and 100-kDa bands were excised from the SDS-PAGE for mass spectrometry analysis in Chinese Academy of Sciences Holdings Co., Ltd. Samples were digested by trypsin at 37 °C overnight and were desalinated using ZipTip. Then the samples were analyzed on the mass spectrometer (5800 MALDI-TOF/TOF, AB SCIEX, USA). Raw data of mass spectrometry was used to identify peptides by searching against the silkworm proteins downloaded from the GenBank database using the software MasCot (version 2.2; Matrix Science, UK).

### 4.6. Immunohistochemistry and Co-Immunoprecipitation (Co-IP)

The MSG and PSG from Day-3 fifth-instar larvae were fixed in 4% paraformaldehyde for 2 h at 4 °C. The samples were embedded in paraffin and 5 μm-thick sections were made for immunofluorescence analyses. The sections were treated with the primary anti-BmOsi9a and anti-Myc, then, with the FITC-labeled secondary goat anti-mouse IgG (Sigma) for anti-Myc and CY3-labeled secondary goat anti-rabbit IgG (Sigma) for anti-BmOsi9a. After washing with phosphate-buffered saline (PBS), the sections were stained with DAPI and observed under a confocal microscope (OLYMPUS FV100, Tokyo, Japan).

The Capturem IP and Co-IP Kit (Takara) was used for Co-IP following the manufacturer’s instructions. Proteins (100 μg) extracted from the MSG were incubated with anti-Ser1 and anti-BmOsi9a antibodies for 1 h at 4 °C. The anti-Ser1 antibody was prepared by the synthesis of Ser1 polypeptides (ChinaPeptides Co., Ltd., Shanghai, China), which were used to immunize rabbits and then to collect serum for purifying the antibody. Rabbit IgG was used as a negative control. The columns of Capturem Protein A were balanced with IP buffer (1.0 M glycine, 2 M NaCl, pH 9.0). The incubation samples were transferred into the columns and centrifuged at 1000× *g* for 1 min. The columns were washed with 100 μL of wash buffer and centrifuged at 1000× *g* for 1 min. Then, 100 μL of elution buffer (0.1 M glycine, pH 2.5) was used for elution, 10 μL of neutralization buffer (1 M Tris, pH 8.5) was pre-installed in the collecting tube, and samples were centrifuged at 1000× *g* for 1 min. Eluted samples were separated by SDS-PAGE following Western blotting with the anti-BmOsi9a antibody.

### 4.7. Mechanical Testing of Silk Fiber

Single silk fibers were obtained and their mechanical properties and strain–stress curves were determined according to previously described methods [21,36]. The single silk fiber was reeled from the cocoons in hot 0.5% (*w*/*v*) NaHCO_3_ solution for 2 min. The average cross-sectional diameter was measured across two brains under a scanning electron microscopy (Imager.A2, ZEISS, Oberkochen, Germany). Single-fiber testing was performed under ambient conditions (24 °C and 60% humidity) using an AG-X plus instrument (Shimadzu, Tokyo, Japan) with a strain rate of 1 mm/min until the fiber broke. For each group, 25 fibers were measured for statistical analyses. Stress–strain curves and mean curves were calculated and drawn using OriginPro 9.0 (OriginLab Corporation, Northampton, MA, USA), including elongation, maximum strength, elastic modulus, and toughness.

### 4.8. ATR-FTIR and XRD Analyses

Single silk fibers were used for attenuated total reflectance Fourier-transform infrared spectroscopy (ATR-FTIR) and X-ray diffraction (XRD) as described previously [17]. ATR-FTIR was performed using a Nicolet iN10 with a Slide-On ATR objective lens (Thermo Scientific). The spectra of silk fibers were measured in the 650–4000 cm^–1^ range at a resolution of 8 cm^–1^ with 256 scans. The applied ATR was set to 75 current pressure. The spectral data were collected and processed using OMNIC v. 9 (Thermo Scientific) and PeakFit v. 4.12, including baseline correction, deconvolution of amide I bands, and peak fitting. XRD was performed using an X’Pert^3^ Powder X-ray diffractometer (PANalytical, Almelo, Netherlands) with Cu Kα radiation from a source operated at 40 kV and 40 mA. Following a previously described method [17], all samples were mounted on aluminum frames and scanned from 5° to 50° (2*θ*) at a speed of 2.0°/min. MDI JADE 6.5 was used to calculate the relative crystallinity, according to the following formula: crystallinity (%) = (X/Y) × 100; where, X is the net area of diffracted peaks and Y is the net area of diffracted peaks + background area [17].

### 4.9. Statistical Analysis

All data results were expressed as mean ± standard deviations. One-way analysis of variance was performed following unpaired two-tailed Student’s *t* test. Statistically significant difference were set at * *p*-value < 0.05, ** *p*-value < 0.01, and *** *p*-value < 0.001.

## Figures and Tables

**Figure 1 ijms-21-01888-f001:**
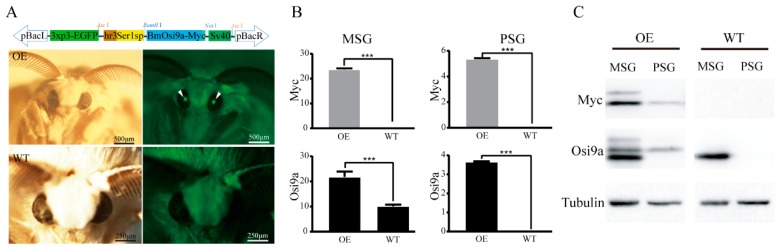
Transgenic overexpression of BmOsiris9a in the silk gland. (**A**). The transgenic expression vector pBac [hSer1sp-BmOsi9a-Sv40, 3xp3-EGFPaf] was constructed to overexpress *BmOsi9a* of *B. mori*. Fluorescent images of a G1 moth of the transgenic overexpression (OE) of BmOsi9a and the wild type (WT). Arrowheads denote the positions of GFP fluorescence. (**B**). Relative mRNA levels of *BmOsi9a* and *Myc* were investigated by qRT-PCR in the middle silk gland (MSG) and the posterior silk gland (PSG) of the OE and the WT. *sw22934* was used as a control (*** *p* < 0.001). **C**. Immunoblot analysis of BmOsi9a and Myc-tag in the MSG and the PSG of the OE and the WT. Proteins dissolved from the MSG and the PSG were separated by SDS-PAGE and immunoblotted with anti-BmOsi9a and anti-Myc antibodies. Tubulin was used as a control.

**Figure 2 ijms-21-01888-f002:**
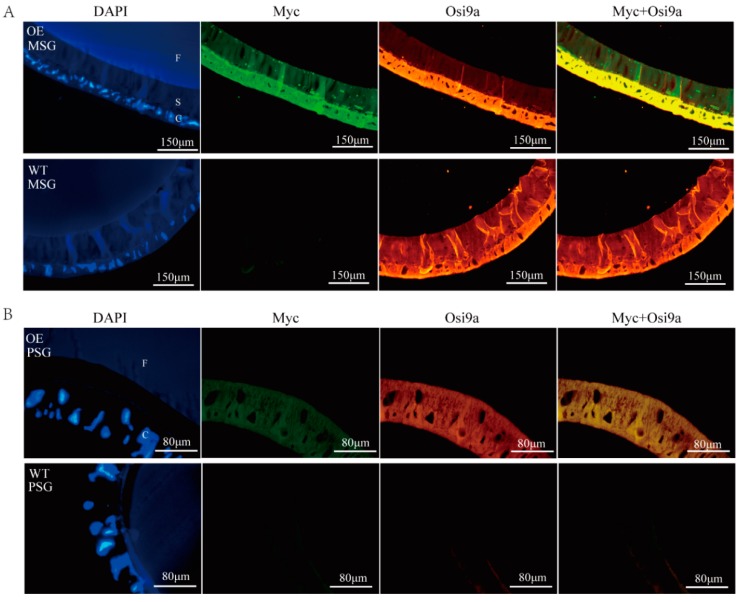
Location of transgenic overexpressed BmOsiris9a in the silk gland. (**A**). Immunofluorescence of BmOsi9a and Myc in the MSG. (**B**). Immunofluorescence of BmOsi9a and Myc in the PSG. Sections were prepared from the MSG and the PSG of the OE and the WT and were observed under ultraviolet light for DAPI, blue light for FITC, and green light for CY3. **C**: Cells; S: Sericin layer; F: Fibroin layer.

**Figure 3 ijms-21-01888-f003:**
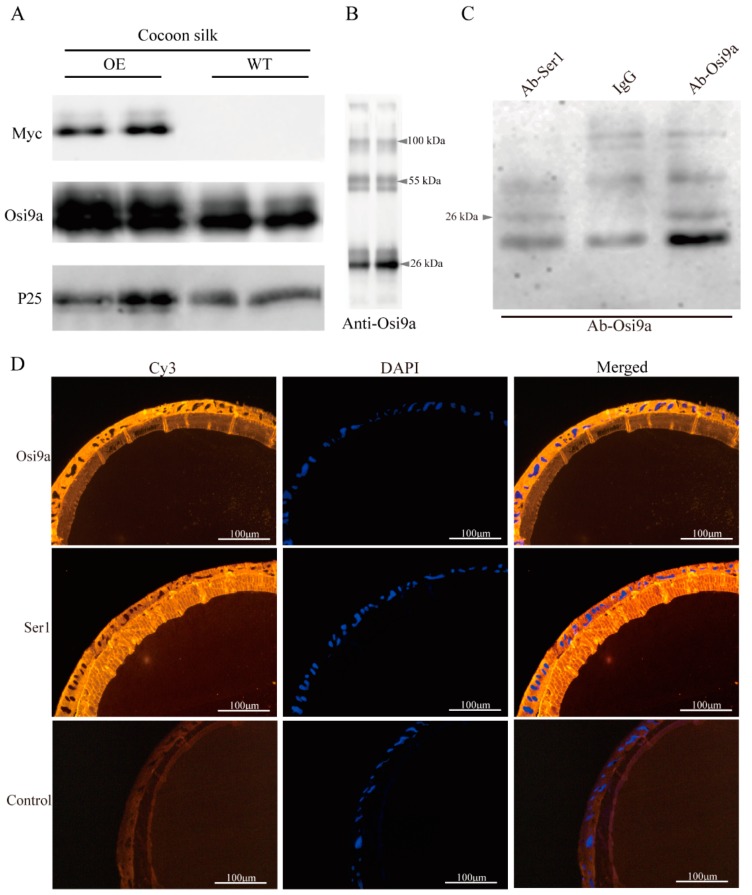
Interaction between BmOsirsi9a and Ser1. (**A**). Western blotting of BmOsi9a and Myc-tag in the cocoon silks of the OE and WT, using P25 as a control. Proteins dissolved from the cocoon were separated by SDS-PAGE and immunoblotted with anti-BmOsi9a, anti-Myc, and anti-P25 antibodies. (**B**). Western blotting of BmOsi9a in the cocoon silks. Two high-molecular-weight bands (55 and 100 kDa) contained the Ser1 protein, as determined by LC-MS/MS analysis. The two lanes are twice biological replicates. (**C**). Co-IP analysis of BmOsi9a and Ser1. Lane 1: proteins of the MSG incubated with anti-Ser1 antibody; Lane 2: proteins of the MSG incubated with rabbit IgG; Lane 3: proteins of the MSG incubated with anti-BmOsi9a antibody. (**D**). Immunofluorescence of BmOsi9a and Ser1 in the MSG. Sections were prepared from the MSG and observed under ultraviolet light for DAPI and green light for CY3.

**Figure 4 ijms-21-01888-f004:**
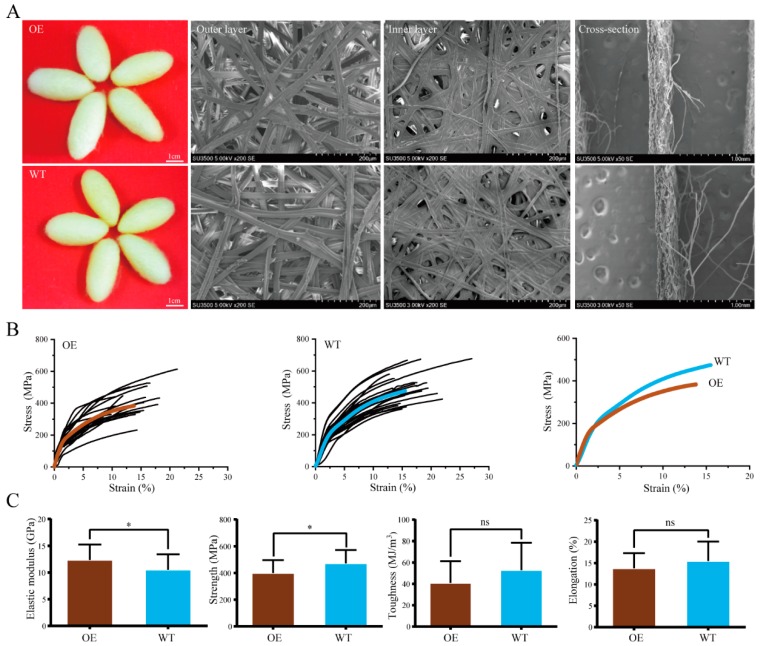
Comparison of phenotypes and mechanical properties of silks from the OE and WT. (**A**). The phenotype of cocoons and silk fibers of the OE and WT. (**B**). Stress–strain curves of the silk fibers. (**C**). Toughness, maximum strength, toughness, and elastic modulus. Error bars, SD; * *p* < 0.05.

**Figure 5 ijms-21-01888-f005:**
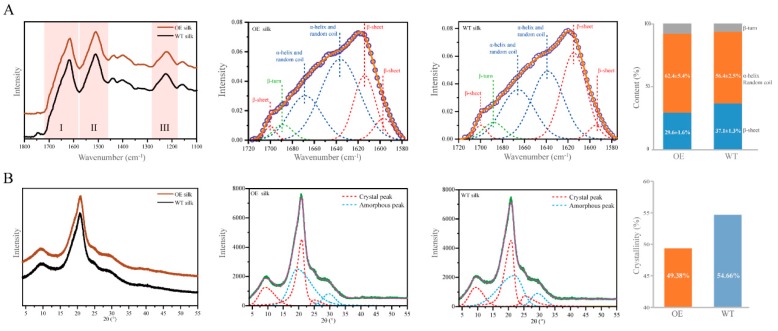
Secondary structural characteristics and crystallinity of silk fibers. (**A**). ATR-FTIR spectra from 1100 to 1800 cm^−1^ of silk from the silks of the OE and WT. Deconvolution of the corresponding amide I band. Comparison of the *β*-sheet content in the silks between the OE and WT. (**B**). X-ray diffraction diffractograms of silk from the silks of the OE and WT. Deconvolution of the silk diffractograms, with intensity as a function of the scattering angle 2*θ*. Comparison of the crystallinity of the silks between the OE and WT.

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
