# Peer review of "Effects of Osiris9a on Silk Properties in Bombyx mori Determined by Transgenic Overexpression"

_ijms, 2020, doi:10.3390/ijms21051888_

Round 1

Reviewer 1 Report

In this study Cheng et al carried out the functional analysis of Osiris9a gene/protein in the silkworm Bombyx mori. Osiris9a is one of the Osiris family genes that are playing multiple biological roles. However, functions of this family genes are still largely unknown. Osiris9a is unique in terms that it is expressed strongly in the middle silk gland of the silkworm. The functional analysis of this gene should be of significance because the silk gland is a tissue unique to the Lepidoptera and also this gene would possibly be involved in the cocoon production, whose functional analysis should contribute to the industrial application of the silkworm. However, I consider this manuscript cannot be acceptable according to the insufficiency of the data as well as its explanation. See the details described below.

  1. Line 122. There is no data for the LC-MS/MS analysis. Interaction between Osiris9a and Ser1 is one of the most important topics in this study and thus the authors should present the detailed data. Also, why the putative Ser1 bands in Fig. 3B is just 55 or 100 kDa? The Ser1 protein size would be larger.
  2. Fig. 3B. Why are there two lanes? Explanation is necessary.
  3. There is no description about the anti-Ser1 and anti-BmOsi9a antibody. If they were generated in this study, show the detailed method. If they were generated in the previous study, show the reference.
  4. For the misexpression experiment. First, describe clearly that the “ser1 promoter” was used in this study. If so, why the transgene expression is occurring also in the PSG? The explanation in line 196 is unclear. I wonder this may be due to the position effect, that is, the transgene expression is affected by the piggyBac insertion position.
  5. Another question concerning this is that if the authors are willing to misexpress osi9a in the PSG, why don’t they use the PSG promoter, such as h-fibroin or l-fibroin, or the ubiquitous promoter, such as A3?

    6. If the authors attempt to elucidate osi9a function, I propose they should do the knockout analysis instead of the misexpression analysis.

Author Response

Thank you very much for pointing these problem and your valuable suggestion. We would like to provide the description and promote them in the revised manuscript.

Reviewer 2 Report

Effects of Osiris9a on Silk Properties in Bombyx mori Determined by Transgenic Overexpression

The authors characterized the functional role of Osiris9a gene on the formation of silk in B. mori. The experiments were well organized and the manuscript was well written.

However, I feel that the scientific meaning of this study is not fully discussed. Thus if possible, it is much better to add scientific novelty and meaning in the discussion part. 

Author Response

Thank you very much for your valuable comments of our research work. Following your suggestion, we have add some discussion to interpret the scientific novelty for this study in the last paragraph of the discussion section.

Round 2

Reviewer 1 Report

I request further following revisions.

  1. Line 64. The authors should explain here that the Ser1 promoter was used to overexpress BmOsi9a in the middle silk gland.
  2. Line 122-123. More detailed information is necessary for the mass spectrometry analysis. How did the authors identify that the excised bands correspond to Ser1? A software like as Mascot was used? I propose to describe in detail in the Materials and method, not in the result.
  3. Line 200. two reason -> two reasons
  4. Line 279. was obtain -> was obtained

Author Response

Point 1: Line 64. The authors should explain here that the Ser1 promoter was used to overexpress BmOsi9a in the middle silk gland.

Response: Following your suggestion, we add some description here in the revised manuscript.

Point 2: Line 122-123. More detailed information is necessary for the mass spectrometry analysis. How did the authors identify that the excised bands correspond to Ser1? A software like as Mascot was used? I propose to describe in detail in the Materials and method, not in the result.

Response: Thank you very much for your suggestion. Following your suggestion, we add a paragraph to describe the mass spectrometry analysis in the materials and method section. The mass spectrometry analysis used the MALDI-TOF/TOF, instead of LC-MS/MS. We are very sorry for the previous description error about the method.

Point 3: Line 200. two reason -> two reasons

Response: Thank you very much for pointing the problem. We have revised it.

Point 4: Line 279. was obtain -> was obtained

Response: We are very sorry for the error and change it in the revised manuscript.